# Multifunctional Membranes Based on β-Glucans and Chitosan Useful in Wound Treatment

**DOI:** 10.3390/membranes12020121

**Published:** 2022-01-20

**Authors:** Sonia Trombino, Federica Curcio, Maria Luisa Di Gioia, Biagio Armentano, Teresa Poerio, Roberta Cassano

**Affiliations:** 1Department of Pharmacy, Health and Nutritional Sciences, University of Calabria, 87036 Rende, Italy; sonia.trombino@unical.it (S.T.); federica.curcio@unical.it (F.C.); ml.digioia@unical.it (M.L.D.G.); biagio.armentano@unical.it (B.A.); 2National Research Council, Institute on Membrane Technology (CNR-ITM), Via Pietro BUCCI, University of Calabria, 87036 Rende, Italy

**Keywords:** chitosan membranes, β-glucans, wounds, antibacterial properties, antioxidant properties

## Abstract

In this work, bio-based membranes prepared using a crosslinked β-glucans–chitosan dispersed in the chitosan matrix useful in promoting wound healing were studied for the first-time. Wound healing is a process that includes sequential steps designed to restore the structure and function of damaged cells and tissue. To minimize damage and the risk of infection during the healing process and to promote restoration of the integrity of damaged tissue, the wound should be dressed. Generally, according to their function in the wound, dressings are classified on the basis of type of material and physical form. The substances used to make a dressing are generally natural polymers such as hydrocolloids, alginates, polyurethane, collagen, chitosan, pectin and hyaluronic acid. The combination of polymeric substances, with antibacterial and antioxidant properties, could be exploited in the biomedical field for the development of biocompatible materials able to act as a barrier between the wound and the external environment, protecting the site from bacterial contamination and promoting healing. To this aim, bio-based membranes were prepared by the phase inversion induced by solvent evaporation, using the crosslinked β-glucans–chitosan obtained by esterification reactions as a functional additive in the chitosan membrane. The reaction intermediates and the final products were characterized by Fourier transform infrared spectroscopy (FTIR) and differential scanning calorimetry (DSC) while the morphological properties of membranes were analyzed using electronic scanning microscopy (SEM). The chemical bonding between chitosan and β-glucans allowed for the obtainment of a better dispersion of the combined new material into the membrane’s matrix and as a consequence, an enhanced antibacterial property evaluated through in vitro tests, with respect to the starting materials.

## 1. Introduction

Wound healing is an important biological process involving tissue repair and regeneration following damage to the skin or intrinsic skin barriers [1,2]. Failure to heal chronic wounds represents an important public health problem, which is characterized by a protracted inflammatory phase, delayed cell proliferation, epigenetic modifications, weak re-epithelialization, and modified angiogenesis. Therefore, the wound management strategy plays a significant role in the outcome of the healing process and in the decrease in a patient’s pain and discomfort. However, wound care is costly, and it is necessary to have wound dressings that are cheap for the increasing population. However, the materials currently used for dressings have several limitations, such as poor antimicrobial effects and an inability to ensure sufficient moisture to enhance the wound healing process [3,4,5,6,7]. Therefore, it is necessary to develop more advanced materials capable of favoring the listed processes. Dressings intended for wound treatment are mostly developed from natural and synthetic polymers [8,9,10]. In this regard, β-glucans, which are 1,3-bonded glucose polymers; 1,4 or 1,6 β-glycosidic bonds, coming from a variety of sources including yeasts, cereals, and mushrooms [11], are increasingly used in dermatology, including in wound care, for their multiple properties [12,13,14,15,16,17]. Through their ability to attract macrophages, neutrophils and other immune cells, these substances fight infections at the wound site and enhance the migration and proliferation of keratinocytes and fibroblasts, which are critical events in the complicated wound healing process [18]. The topical application of β-glucans is on the rise due to their antioxidant, anti-inflammatory and regenerative activities, which could help as an additional therapy in the treatment of various skin diseases [19,20,21]. Furthermore, some β-glucans also possess anti-infective properties and exhibit antibacterial activity against a broad spectrum of Gram-positive and Gram-negative bacteria [22,23]. Conversely, chitosan, a natural linear polysaccharide composed of D-glucosamine and N-acetyl-D-glucosamine, linked via β (1–4) bonds derived from the deacetylation of chitin [24], has several properties such as antibacterial activity, good biodegradation, exceptional biocompatibility, and non-toxicity that make it an excellent candidate as a base material in the production of wound treatment materials [25,26,27]. However, the antibacterial activity of chitosan is highly dependent on its solubility. To date, the common methods to improve the antibacterial activity of chitosan are chemical modification of chitosan and the combination of chitosan with other antimicrobial materials. Chitosan is a versatile material that can be used in various forms, such as beads, membranes, coatings, fibers, and sponges [28]. Chitosan-based membranes have been extensively studied as wound dressings due to their easy production and long shelf life [29]. In this work, novel chitosan-based membranes, in which the synthesized material obtained through esterification between β-glucans carboxylated and chitosan are dispersed, were prepared by the phase inversion induced by solvent evaporation. The improved performance, with respect to the individual biopolymers in terms of antibacterial and antioxidant properties, were evaluated by carrying out in vitro studies of prepared membranes.

## 2. Materials and Methods

### 2.1. Materials

All chemicals used in this work were commercially available and were used as received without further purification. β-D-glucans mixture (β-1,3 and β-1,6) was extracted from Saccharomyces Cerevisiae and was purchased from Farmalabor Srl (Canosa di Puglia, Italy), chitosan (Mw 327 kDa, Deacetylation ≥ 75%), orthophosphoric acid (H_3_PO_4_), sodium nitrite (NaNO_2_), formic acid (HCOOH), dimethylacetamide (DMAc), lithium chloride (LiCl), azodicarboxylate-dipiperidine (ADDP), tributylphosphine (Bu_3_P), hydrochloric acid (HCl), sodium hydroxide (NaOH), and borate buffer pH 8.5 (0.1 M boric acid/NaOH 1 M) were purchased from Sigma Albrich (Milan, Italy).

Acetone, ethyl ether, ethanol, and methanol were purchased from Fluka Chemika-Biochemika (Buchs, Switzerland) and Carlo Erba Reagenti (Milan, Italy). Phenolphthalein salt, methyl red, and methylene blue were purchased from Carlo Erba Reagenti (Milan, Italy). Glacial acetic acid was purchased from VWR Prolabo Chemicals (Milan, Italy).

For antibacterial activity, strains of *Escherichia coli* (ATCC 25922), *Klebisella pneumoniae* (ATCC 13883), *Pseudomonas aeruginosa* (ATCC 27853), and *Staphylococcus aureus* MSSA (ATCC 25923), provided by Remel Microbiology (Thermo Fisher, Waltham, MA, USA), were used. Cells were cultured in Mueller–Hinton II broth (MHB; Difco, Detroit, MI, USA) containing 2 g/L beef infusion solids, 17.5 g/L casein hydrolysate, and 1.5 g/L starch.

### 2.2. Instrumentation

FTIR spectra were recorded by a Jasco 4200 spectrophotometer using potassium bromide (KBr) foils (or powder) provided by Sigma-Aldrich (Saint Louis, MO, USA). UV–Vis spectra were recorded by Jasco V-530 UV/Vis spectrophotometer using 1 cm thick quartz cells. Differential scanning calorimetry (DSC) was performed with the NETZSCH DSC 200 PC instrument. Samples were freeze-dried by “freezing–drying” Micro Modulyo, Edwards. The membrane surface wettability was evaluated by water contact angle (CA) measurements using a CAM 200 device (KSV Instruments, Ltd., Helsinki, Finland). SEM images were performed by FEI QUANTA 200, FEG equipped with an Oxford Inca 300 EDX system. Magnifications were made in a range of 20–100 µm.

### 2.3. Synthesis of Carboxylated β-Glucans

To promote the binding between β-glucans and biologically active substances, it was necessary to perform the β-glucans carboxylation reaction [30,31] using the reagents reported in Table 1 and the procedure described below.

In a 1 L three-neck flask equipped with a drip funnel and magnetic stirrer and flamed under nitrogen, β-glucans and H_3_PO_4_ were added. After 1 h and 30 min, one-third of the total amount of sodium nitrite was added. The solution was kept under vigorous stirring for 5 min. It was left for 1 h and 45 min without agitation and a stable and compact foam was formed. Next, another portion of sodium nitrite was added, under stirring and then left without agitation for 45 min. A third addition was made by repeating the same procedure. After 45 min, 85% formic acid was added to neutralize the excess sodium nitrite. The product was precipitated with acetone and ice-cold ethyl ether (exothermic reaction). Subsequently, the entire content of the flask was filtered, the resulting product was washed with distilled water, and further washes were performed with ethanol. The obtained product was vacuum dried and characterized by FTIR and DSC.

### 2.4. Determination of the Carboxyl Groups Content of Carboxylated β-Glucans

The sample for carboxyl groups determination was prepared by suspending 0.05 g of carboxylated β-glucans mixture into a mixture consisting of 2.5 mL of borate buffer (pH 8.5) and 2.5 mL of an aqueous solution of methylene blue (300 mg/L), which was left under stirring for 1 h at room temperature [32,33]. Subsequently, the suspension was filtered, 1 mL of the filtrate was taken, acidified with 1 mL of 0.1 N HCl, and 8 mL of distilled H_2_O was added. The methylene blue content of this solution was determined through a UV–Vis spectrophotometer by employing a calibration line relative to methylene blue (ε = 86,126 mol^−1^Lcm^−1^). The method is based on binding of the free methylene blue cation to carboxyl functions and subsequent determination of the decrease in its concentration in the solution. The resulting amount of unabsorbed free methylene blue was calculated and used in Equation (1) to obtain the content of carboxyl groups, which was of 0.430 mmol.
(1)$$\mathrm{mmol}\;\mathrm{COOH}/g\;\mathrm{dry}\;\mathrm{sample}\;=\frac{\left(7.5-A\right)\times 0.00313}{E}$$
where *A* is the total amount of free methylene blue in milligrams and *E* is the weight of the dry sample in grams.

### 2.5. Synthesis of β-Glucans–Chitosan

The synthesis of β-glucans–chitosan was carried out by using the Mitsunobu reaction that allows esters with high yields to be obtained from a primary or secondary alcohol and a carboxylic acid using a trisubstituted phosphine and a disubstituted azodicarboxylate [34].

The experimental procedure involves the suspension of carboxylated β-glucans in DMAc and LiCl within a 500 mL coded flask equipped with a CaCl_2_ valve and maintained in a nitrogen current, under stirring and cold for 60 min. ADDP, tributylphosphine, and chitosan were then quickly added (Table 2). The reaction was conducted at room temperature for 24 h and under nitrogen atmosphere. The obtained product was washed with hot methanol, filtered, dried under vacuum, and was finally characterized by FTIR and DSC. The product obtained is in the form of a white powder.

### 2.6. Determination of the Degree of Substitution (DS) by Volumetric Analysis

To calculate the degree of substitution, he β-glucans–chitosan mixture was subjected to basic hydrolysis (Table 3) [35,36]. A small amount of product (0.05 g) was dissolved in an ethanolic solution of NaOH 0.25 M. The mixture was kept under stirring at 100 °C for 24 h. Subsequently, titration with 0.1 N HCl was performed, using phenolphthalein as pH indicators for the first equivalence point and methyl red for the second.

Excess NaOH was titrated at the first equivalence point (eq. p.), while the second equivalence point indicates neutralization of the salt of the acid present. The moles of HCl between the first and second equivalence points correspond to the moles of free ester.

DS is determined by the following in Equation (2):(2)$$\mathrm{DS}\;=\frac{MM\;glucosidic\;unit}{\frac{g\;sample\;}{n\;free\;ester}-MM\;free\;ester-MM\;H_{2}O}$$
where, *MM glucosidic unit* is the molecular mass of one unit of glucose. *n free ester* is the number of moles of hydrolyzed ester (V2 eq. p.–V1 eq. p.); *g sample* is the weight of the sample (0.05 g); *MM free ester* is the molecular weight of the free ester; *MM H*_2_*O* is the molecular weight of one H_2_O molecule (18 g/mol).

### 2.7. Membrane Preparation Procedure

Membranes made of pure chitosan © were prepared by using the evaporation-induced phase separation (EIPS) technique [37]. Polymer solution was prepared by dissolving 3 *w*/*v* % of C in 1% (*v*/*v*) acetic acid aqueous solution and stirred until a uniform solution was obtained. Then, the solution was poured onto a Petri dish and dried at 60 °C for 24 h in an oven; after that, the formed membrane was peeled off from the Petri dish and was washed with distilled water. Membranes made of chitosan and β-glucans and chitosan and β-glucans–chitosan were prepared following the same procedure before described by adding 0.1 g of β-glucans or β-glucans–chitosan to the polymer solution of chitosan.

Table 4 summarizes the composition of prepared membranes, and their name used in the paper.

### 2.8. In Vitro Tests of Antibiotic Activity on Plate and on Broth

The antibacterial activity of the membranes C, C-β-glucans and C-β-glucans–chitosan against the bacterial strains *Escherichia coli*, *Pseudomonas aeruginosa*, *Staphylococcus aureus* and *Klebsiella pneumonia* were evaluated [38,39,40]. The susceptibility of bacteria to our membranes was evaluated by the Kirby–Bauer disc-diffusion method performed according to CLSI guidelines, and the results were interpreted using CLSI breakpoints [41,42,43,44,45].

Disks, with a diameter of 6 mm, of all the prepared membranes were used for the evaluation of antimicrobial activity; blank antimicrobial susceptibility disks (Oxoid™) were used as negative control. Specifically, cultures of tested bacteria were adjusted to a turbidity of 0.5 McFarland standard (10^6^ CFU/mL) before inoculation on agar plates with sterile cotton swabs. A cotton swab soaked in cell culture was streaked over one surface of the agar plate to obtain an even layer of bacteria over the entire surface. After 10 min, new membrane discs and the control disc were placed on the inoculated surface of the agar plates; then, all plates were incubated at 37 °C overnight. After 24 h, diameters of inhibition were measured, and the susceptibility of the membranes toward the bacteria was expressed in terms of resistance (R), moderate susceptibility (I) and susceptibility (S). The result obtained on the individual bacterial strains was confirmed in terms of mm (zone of inhibition). The sensitivity test was repeated three times independently.

The growth of bacteria was also evaluated on broth according to CLSI guidelines [45,46,47].

Briefly, microorganisms prepared from bacterial cultures in Mueller–Hinton medium (MHB) were added to suspensions of the microorganisms at a concentration of 106 CFU/mL for each sample in a 1:1 ratio. Growth or lack thereof of microorganisms was determined after incubation for 24 h at 37 °C by turbidimetry (600 nm wavelength) [48,49]. MHB was prepared by dissolving 4.2 g of powdered medium in 200 mL of distilled water; the solution was brought to a boil at 100 °C for approximately 1–2 min and then cooled completely.

The absorbance of the samples was measured for each solution after 15, 24, 39, 46, and 111 h.

### 2.9. Evaluation of Antioxidant Capacity

Antioxidant capacity was examined in rat liver microsomal membranes. These membranes consist of phospholipids with a high content of polyunsaturated fatty acids and represent the ideal substrate of the lipid peroxidation process. During this reaction, fatty acids are transformed into toxic metabolites such as aldehydes. Malondialdehyde is consistently generated and is a good indicator of the rate of peroxidation. To mimic the process of lipid peroxidation, a pro-oxidant agent, such as *tert*-butylhydroperoxide (*t*-BOOH), was employed, which catalyzes the formation of hydroxyl radicals (-OH^●^) responsible for peroxidation [50].

Rat liver microsomes were prepared from Wistar rats by tissue homogenization with 5 volumes of a fresh 0.25 M sucrose solution containing 5 mM HEPES, 0.5 mM EDTA, pH 7.5, in a Potter–Elvehjem homogenizer. Microsomal membranes were isolated by removal of the nuclear fraction at 8000 rpm for 10 min and the mitochondrial fraction at 18,000 rpm for 10 min. The microsomal fraction was sedimented at 10,500 rpm for 60 min and was washed once with 0.15 M HCl, collected, and centrifuged again at 10,500 rpm for 30 min. Membranes were suspended in 0.1 M phosphate buffer pH 7.4 and stored at −80 °C. Protein concentration was determined by the Bradford method using the Bio.Rad assay [51].

Microsomes were suspended in 0.1 M phosphate buffered saline (PBS) at pH 7.4. Predetermined amounts of the functionalized β-glucan-chitosan membrane were quickly added to the microsomes. Subsequently, a 0.01 M *tert*-BOOH solution was added. The suspensions, obtained by homogenization, were incubated at 37 °C under stirring, in air and in the dark.

One milliliter of microsomal suspension (0.5 mg protein) was added to a solution consisting of 3 mL of 0.5% trichloroacetic acid (TCA), 0.5 mL of thiobarbituric acid (TBA), and 0.07 mL of 0.2% hydroxytoluenbutylate (BHT) in 95% ethanol. The obtained samples were incubated in a bath at 80 °C for 30 min and then centrifuged.

After incubation, the TBA-MDA complex (pink color chromogen) was detected by UV–Vis spectrophotometry at 535 nm. Results are expressed as nmol of MDA/mg of protein in the lipid sample.

## 3. Results and Discussion

The mixture of β-D-glucans (β-1,3 and β-1,6) and chitosan, substances with antibacterial and antioxidant activity, were covalently linked by esterification reaction following the Figure 1.

Initially, the mixture of β-D-glucans (β-1,3 and β-1,6) was subjected to a carboxylation reaction and subsequently characterized. Quantitative analysis of carboxyl groups [32,33] revealed a content of 0.430 mmol per 50 mg of β-glucans mixture. From the FTIR spectrum (Figure 1), the presence of a new band at 1715 cm^−1^ due to the stretching vibration of the C=O of the acid was revealed. After carboxylation, the β-glucan mixture was esterified with chitosan and characterized by FTIR and DSC. From the data obtained, the presence of a new band at 1725 cm^−1^ can be attributed to the C=O stretching vibration of the ester group. For further confirmation of esterification, quantitative analysis of the ester groups was performed by determining, by volumetric analysis, the degree of substitution (DS), which was found to be 0.820. DSC curves of the β-glucans mixture (curve c), the carboxylated β-glucans mixture (curve b), and β-glucans–chitosan (curve a) are shown in Figure 2, indicating an endothermic peak at 228, 225, and 242 °C, respectively. The shift at a higher temperature of the latter peak further confirms the formation of ester bond between β-glucans and chitosan.

The obtained membranes were morphologically characterized by scanning electron microscopy (SEM) (Figure 3). From these images, it is possible to observe how the chitosan crosslinked with β-glucans permitted a better dispersion of β-glucans into a chitosan matrix allowing a quite smooth surface. A rough surface was instead evidenced in the case in which only β-glucans were dispersed in the polymer matrix.

The rougher surface generally causes a hydrophobic surface [52] as also evidenced by contact angle measurements of C-β-glucans membranes (Figure 4), indicating a slight surface hydrophobicity with a contact angle value of 98° with respect to Chitosan and C-β-glucans–chitosan membranes that are respectively 88° and 89.5°. Surface proprieties, in terms of roughness and hydrophilicity/hydrophobicity, affect antimicrobial activity, especially the initial interaction. An increase in hydrophilicity, enhancing the antiadhesive property, was found to inhibit bacterial colonization. On a highly hydrophilic membrane surface, a film of water molecules will adhere to the membrane surface, thus preventing bacteria from attaching to the membrane surface.

There are many factors other than surface properties that impact the antibacterial properties of chitosan, such as type of microorganism, Mw of chitosan, deacetylation degree, concentration, pH, source of chitosan, temperature, phase of cell growth, and chitosan derivatives. All these factors make it difficult to understand the antibacterial mechanism of chitosan [26,53].

The antimicrobial activity was measured against Gram-positive (Staphylococcus aureus) and Gram-negative bacterial strains (*Escherichia coli*, *Pseudomonas aeruginosa* and *Klebsiella pneumoniae*) by using a disc-diffusion susceptibility test (Figure 5, Kirby–Bauer test). The obtained results, under the specified operating conditions, are summarized in Table 5 and indicate that the chitosan membrane and C-β-glucan-chitosan membrane do not possess antibacterial activity, but only the β-glucan-chitosan membrane exhibits this activity against *Pseudomonas aeruginosa* (S), *Klebsiella pneumonia* (S) and *E. coli* (I), while *Staphylococcus aureus* appears to be resistant (R).

Moreover, the sensitivity of bacteria to the different polysaccharides was also confirmed by monitoring their growth in broth for 24 h. Bacterial growth was assessed as cell density increase (Abs) in presence of membrane discs of chitosan, C-β-glucans and C-β-glucans–chitosan, and it was compared to the control, i.e., growth of the bacterium without treatment (Figure 6). Significant results were obtained when β-glucans–chitosan membranes treatment in *P. aeruginosa*, *E. coli* and *K. pneumoniae* (*p* value = 0.0005), while S. aureus growth was unaffected by β-glucans–chitosan exposure (*p* value = 0.05).

The ability of chitosan, C-β-glucans and C-β-glucans–chitosan membranes to inhibit lipid peroxidation, induced by a free radical generator such as tert-BOOH, was examined in rat liver microsomal membranes over 120 min of incubation. The ability of the synthesized materials to inhibit lipid peroxidation was shown to be time dependent. Results indicated that C-β-glucans and C-β-glucans–chitosan membranes showed the same antioxidant capacity of the chitosan membranes, with a slight protecting effect of microsomes from tert-BOOH-induced lipid peroxidation (Figure 7).

## 4. Conclusions

Multifunctional membranes with antibacterial and antioxidant activity were prepared by combining the properties of two biomaterials such as β-glucans and chitosan. Initially, the mixture of β-D-glucans (β-1–3 and β-1–6) was carboxylated and then esterified with chitosan to achieve better dispersion in the chitosan membrane matrix. The resulting product, characterized by FTIR and DSC and degree of substitution, was used as new bio-based material for the preparation of multifunctional membranes. The prepared membranes revealed good antibiotic activity of biopolymer and its ability to prevent the growth of microorganisms, in particular, of *E. coli*, *Pseudomonas aeruginosa* and *Klebsiella pneumoniae.* In addition, these membranes have shown the maintenance of the antioxidant activity of the starting materials. The obtained results suggest the possibility of using these combined materials to improve their individual properties, and they may be useful for biomedical applications such as wound treatment patches.

## Data Availability

Data is contained within the article.

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
