# Peer review of "Multifunctional Membranes Based on β-Glucans and Chitosan Useful in Wound Treatment"

_membranes, 2022, doi:10.3390/membranes12020121_

Round 1
Reviewer 1 Report
Reference to the evaluation of the manuscript membranes-1556961, entitled “Multifunctional membranes based on β-glucans and chitosan useful in wounds treatment” The authors has developed multifunctional membranes based on β-glucans and chitosan, I have critically evaluated the manuscript and have following comments.
- Abstract line 17-18 The substances used to make a dressing are generally natural
polymers such as hydrocolloids, alginates, hydrogels.
Hydrogel is not a polymer it is drug delivery system.
- A lot of Literature is available on anti-bacterial activity and anti-oxidant activity of chitosan, but current paper is devoid of such literature in introduction section.
- Chitosan molecular weight is not mentioned?
- Refine quality of all tables. Consult good papers.
- 7 in pure chitosan membrane synthesis no plasticizer is used which make membrane brittle and difficult to peel off from petri plate. How u justify flexibility of membranes?
- Chitosan membrane solution was dried at 60 áµ’c this high temperature over dried membrane sometime and make it brittle and hard.
- Table 4. Composition of prepared membrane
Table contains polymer without polymer ratios/ concentration (gm, mg, % etc). Reconstruct table see good papers.
- Number of studies reported on antibacterial activity of chitosan in current study chitosan shows no antibacterial activity, justify reason with valid reference in discussion section.
- Label FTIR graph.
- Label DSC curves.
- The authors can cite the following articles for discussion part improvement.
Bioinspired sodium alginate based thermosensitive hydrogel membranes for accelerated wound healing - ScienceDirect
Self-crosslinked chitosan/κ-carrageenan-based biomimetic membranes to combat diabetic burn wound infections - ScienceDirect
Author Response
Please find our detailed point-by-point responses to the reviewers’ concerns on our manuscript copied below
Abstract line 17-18 The substances used to make a dressing are generally naturalpolymers such as hydrocolloids, alginates, hydrogels. Hydrogel is not a polymer it is drug delivery system.
Thank you for pointed out this mistake in the abstract section. The authors eliminate the word hydrogel
- A lot of Literature is available on anti-bacterial activity and anti-oxidant activity of chitosan, but current paper is devoid of such literature in introduction section.
The authors added some literature based on antibacterial and anti-oxidant activity of chitosan and the following sentences:
....., has several properties such as antibacterial activity, good biodegradation, exceptional biocompatibility, and non-toxicity that make it an excellent candidate as a base material in the production of wound treatment materials [23-25]. However, the antibacterial activity of chitosan is highly dependent on its solubility. To date, the common methods to improve the antibacterial activity of chitosan are chemical modification of chitosan and combination of chitosan with other antimicrobial materials. Chitosan is a versatile material that can be used in various forms, such as beads, membranes, coatings, fibers, and sponges [26].
- Chitosan molecular weight is not mentioned?
Molecular weight and deacetylation degree of chitosan were added
Mw 327 kDa, Deacetylation ≥ 75 %
- Refine quality of all tables. Consult good papers.
All tables are refined
- in pure chitosan membrane synthesis no plasticizer is used which make membrane brittle and difficult to peel off from petri plate. How u justify flexibility of membranes?
All membranes were prepared without the use of plasticizer, the presence of moisture as in the case of skin contact make them become more flexible.
- Chitosan membrane solution was dried at 60 áµ’c this high temperature over dried membrane sometime and make it brittle and hard.
This temperature was used to accelerate the evaporation of water in order to form membranes faster.
- Table 4. Composition of prepared membrane. Table contains polymer without polymer ratios/ concentration (gm, mg, % etc). Reconstruct table see good papers.
The authors specify the weight % of the β-glucans and β-glucans-chitosan resulted in the chitosan matrix.
- Number of studies reported on antibacterial activity of chitosan in current study chitosan shows no antibacterial activity, justify reason with valid reference in discussion section.
Thank you for the suggestion, the authors added the following sentence with related reference in results and discussion
There are many factors that impact the antibacterial properties of chitosan such as type of microorganism, Mw of chitosan, deacetylation degree, concentration, pH, source of chitosan, temperature, phase of cell growth, chitosan derivatives. All these factors make difficult to understand the antibacterial mechanism of chitosan [26, 53,54].
Zheng, L.-Y.; Zhu, J.-F.; Study on antimicrobial activity of chitosan with different molecular weights, Carbohydr. Polym. 2003, 54, 527–530.
Kong, M.; Chen, X.G.; Xing, K.; Park, H.J. Antimicrobial properties of chitosan and
mode of action: a state of the art review, Int. J. Food Microbiol. 2010, 144, 51–63.
- Label FTIR graph.
The label was added
- Label DSC curves.
The label was added
- The authors can cite the following articles for discussion part improvement.
- Bioinspired sodium alginate based thermosensitive hydrogel membranes for accelerated wound healing - ScienceDirect
- Self-crosslinked chitosan/κ-carrageenan-based biomimetic membranes to combat diabetic burn wound infections - ScienceDirect
The suggested papers have been added in the introduction section
Khaliq, T.; Sohail, M.; Minhas, M. U.; Shah, S. A.; Jabeen, N.; Khan, S.; Hussain, Z; Mahmood, A.; Kousar, M.; Rashid H. Self-crosslinked chitosan/κ-carrageenan-based biomimetic membranes to combat diabetic burn wound infections, Int. J. Biol Macromol, 2022, 197, 157-168.
Abbasi, A.R.; Sohail, M.; Minhas, M. U; Khaliq, T.; Kousar, M.; Khan, S.; Hussain, Z.; Munir, A. A.Bioinspired sodium alginate based thermosensitive hydrogel membranes for accelerated wound healing, Int. J. Biol. Macromol. 2020 155, 751-765.

Reviewer 2 Report
This manuscript is well written. Introduction with adequate context. Methods include all the steps. Results and discussion have scientific soundness. The outcome is supported by the findings.
This is an interesting topic, the authors to investigate bio-based membranes prepared using a crosslinked β-glucans-chitosan dis12 persed in the chitosan matrix, that will be useful to promote wound healing. The well-documented introduction in the literature, but I have a suggestion, at page 2, line 61-61, the phrase "extensively studied as wound dressings due to their easy production and long shelf life" and showed only reference. Please add a sentence how is the relevance of investigating multifunctional membranes based on β-glucans and Chitosan? In materials section, all steps have been written. The findings and discussions are consistent with the dates. Conclusions are supported, these dates.
Author Response
Please find our detailed point-by-point responses to the reviewers’ concerns on our manuscript copied below.
The authors added the following sentence
To date, the common methods to improve the antibacterial activity of chitosan are chemical modification of chitosan and combination of chitosan with other antimicrobial materials.
and the following references:
- Lia J.; Zhuang S. Antibacterial activity of chitosan and its derivatives and their interaction mechanism with bacteria: Current state and perspectives, European Polymer Journal 2020 138, 109984.
- Kong, N.¸ Chen, X.G.; Xing, K.; Park, H.J. Antimicrobial properties of chitosan and mode of action: a state of the art review. Int. J. Food Microbiol. 2010 144, 51–63.
Ma, Y.; Xin L.; Tan, H.; Fan, M.; Li, J.; Jia, Ya.; Ling, Z.; Cheb, Y.; Hu X. Chitosan membrane dressings toughened by glycerol to load antibacterial drugs for wound healing Materials Science & Engineering C 2017, 81, 522–531.